# Targeting fidelity of adenine and cytosine base editors in mouse embryos

Hye Kyung Lee [1], Michaela Willi [1], Shannon M. Miller[2,3,4], Sojung Kim[1], Chengyu Liu[5], David R. Liu[2,3,4] & Lothar Hennighausen[1]

Base editing directly converts a target base pair into a different base pair in the genome of living cells without introducing double-stranded DNA breaks. While cytosine base editors (CBE) and adenine base editors (ABE) are used to install and correct point mutations in a wide range of organisms, the extent and distribution of off-target edits in mammalian embryos have not been studied in detail. We analyze on-target and proximal off-target editing at 13 loci by a variety of CBEs and ABE in more than 430 alleles generated from mouse zygotic injections using newly generated and published sequencing data. ABE predominantly generates anticipated A•T-to-G•C edits. Among CBEs, SaBE3 and BE4, result in the highest frequencies of anticipated C•G-to-T•A products relative to editing byproducts. Together, these findings highlight the remarkable fidelity of ABE in mouse embryos and identify preferred CBE variants when fidelity in vivo is critical.

[1] Laboratory of Genetics and Physiology, National Institute of Diabetes and Digestive and Kidney Diseases, US National Institutes of Health, Bethesda, MD 20892, USA. [2] Merkin Institute of Transformative Technologies in Healthcare, Broad Institute of MIT and Harvard, Cambridge, MA 02141, USA. [3] Howard Hughes Medical Institute, Harvard University, Cambridge, MA 02138, USA. [4] Department of Chemistry and Chemical Biology, Harvard University, Cambridge, MA 02138, USA. [5] Transgenic Core, National Heart, Lung, and Blood Institute, US National Institutes of Health, Bethesda, MD 20892, USA. Correspondence and requests for materials should be addressed to H.K.L. (email: hyekyung.lee@nih.gov) or to L.H. (email: lotharh@niddk.nih.gov)

Clustered regularly interspaced short palindromic repeat (CRISPR)-Cas9 genome editing[1,2] has been widely used to disrupt genomic sites[3–6]. However, its application in reliably and efficiently introducing defined mutations into the mouse genome might be limited as double-stranded DNA breaks created by Cas9 routinely result in non-homologous end joining (NHEJ) repair and the insertion and deletion (indel) of sequences at target sites[7,8]. Base editing[9,10], a newer form of genome editing, directly converts target C•G base pairs to T•A, or target A•T base pairs to G•C, without inducing double-stranded DNA breaks[11], which should make it less likely to cause undesired mutations, such as deletions or insertions. Beyond installing the desired mutations at target nucleotide(s) within the ~5 bp base editing window ('on-target editing'), base editors can, in principle, cause mutations elsewhere within the protospacer ('bystander editing'), near the protospacer ('proximal off-target editing'), or away from the protospacer ('distal off-target editing')[9,10,12–17]. Although cytosine base editors (CBE) and adenine base editors ABE 7.10 (hereafter referred to as ABE) have been used to introduce mutations into mice[14,18–21], the extent and distribution of off-target base edits in the mouse genome have not been analyzed in depth, especially for ABEs[19,21].

Here, we generate mutant mice with a variety of CBEs and ABE and characterize on-target editing and some classes of off-target editing in depth. We also analyze data from published mouse embryo base editing studies to date[14,18,19,21]. ABE almost exclusively generates A•T-to-G•C edits ('anticipated products'), while different generations of CBEs create different degrees of unanticipated products beyond the desired C•G-to-T•A mutations.

Our analysis reveals the remarkable fidelity of ABE use in mouse embryos and identifies preferred CBE variants when fidelity in vivo is paramount, findings important to the use of base editing in mouse genetics in basic and translational research.

## Results

**Targeting fidelity of base editors.** We analyzed the editing location and editing product purity of mutations introduced by five different CBEs (BE2[9,14], BE3[9,18], saBE3[21,22], VQR-BE3[22,] and BE4[23]) and by ABE[10,19,21] in the mouse germline (Fig. 1). In total, we investigated mutations in 436 mutant alleles - 222 alleles edited by CBE variants, and 214 alleles edited by ABE and performed statistical analysis (Supplementary Fig. 1). We use the following nomenclature to describe the location of base edits (Fig. 1a). 'On-target editing': mutations within the expected ~5 bp base editing window. 'Bystander editing': mutations outside this window but within the protospacer. 'Proximal off-target editing': mutations outside the protospacer, but within 200 bp upstream or downstream of the protospacer. This study does not analyze off-target edits away from the protospacer ('distal off-target editing'). To describe the product distribution of base editing, we use the following terms: 'Anticipated products' are C•G-to-T•A for CBEs, and A•T-to-G•C for ABE. 'Unanticipated products' include mutations of Cs or As on the non-targeted strand ('opposite-strand editing'), the conversion of the target C•G or A•T to other base pairs than T•A or G•C, respectively ('unanticipated mutations'), and any indels (Fig. 1c). Thus, base editors resulting in high product purity maximize the production of anticipated

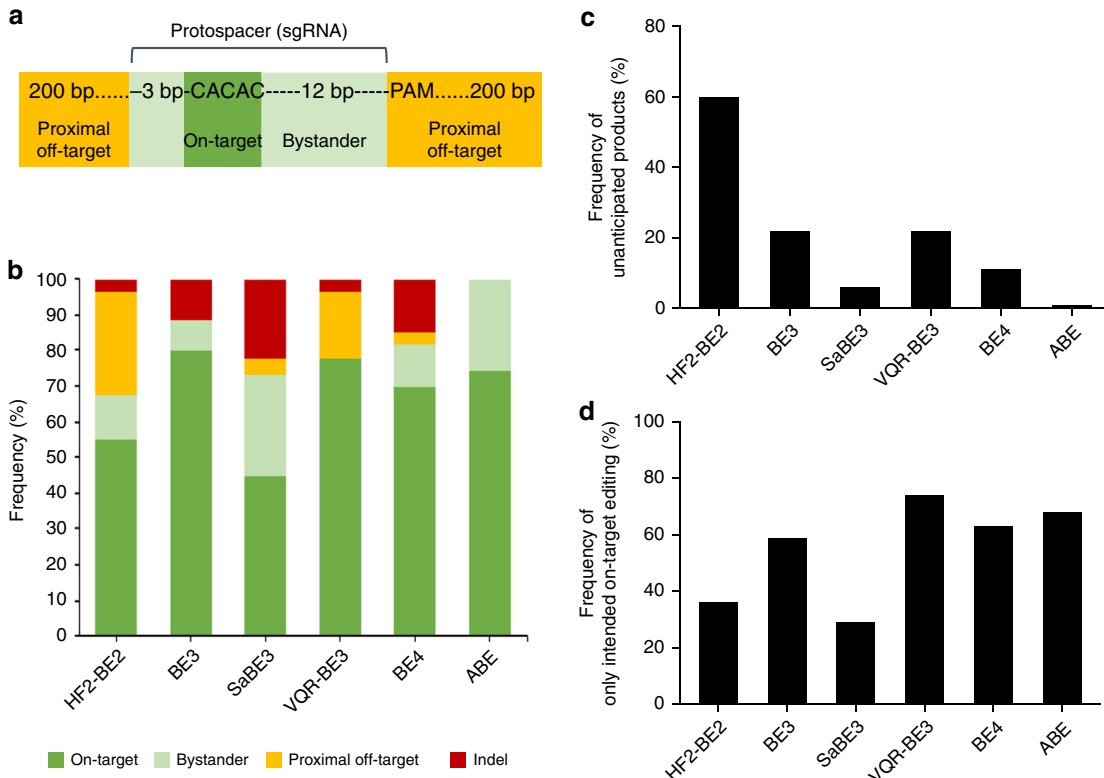

**Fig. 1** Base editing frequency and accuracy. **a** Diagram of a generic target locus with different categories of editing outcomes. On-target (dark green): the target nucleotide within the 5-bp editing window; bystander (light green): a non-target nucleotide within the protospacer; proximal off-target (yellow): outside the protospacer but within 200 bp of the target site. **b** Frequency of alleles carrying on-target mutation, bystander mutation, proximal off-target mutation, and indels. Note that a single allele can carry more than one mutation. The total number of mutant alleles obtained with each base editor was set as 100% (HF2-BE2, 58; BE3, 96; SaBE3, 45; VQR-BE3, 27; BE4, 33; and ABE, 263). The frequency of each category is shown and number of edited alleles for each group and each base editor are shown in Table 1. **c** Frequency of alleles carrying unanticipated products (any mutation different from anticipated products: C•G-to-T•A for CBE, and A•T-to-G•C for ABE). **d** Frequency of alleles that carry only intended (anticipated) edits within the on-target region

**Table 1 Summary of cytosine and adenine base editing results**

| Edited alleles | | CBE | | | | | ABE | | |
|---|---|---|---|---|---|---|---|---|---|
| | | HF2-BE2 | BE3 | SaBE3 | VQR-BE3 | BE4 | ABE[a] | ABE[b] | ABE[c] |
| # of alleles (loci) | | 47 (2) | 91 (2) | 34 (1) | 23 (1) | 27 (1) | 8 (2) | 39 (1) | 167 (3) |
| Editing location | Editing purity | | | | | | | | |
| On-target (5bp window) | # of alleles | 32* (68%) | 77* (85%) | 20 (59%) | 21 (91%) | 23* (85%) | 8* (100%) | 39* (100%) | 148* (89%) |
| | Anticipated products | 21 | 64 | 20 | 21 | 23 | 8 | 39 | 148 |
| | Unanticipated products | 11 | 18 | 0 | 0 | 2 | 0 | 0 | 3 |
| Bystander (within protospacer) | # of alleles | 7* (15%) | 8* (9%) | 13* (38%) | 0 | 4* (15%) | 2* (25%) | 10 (26%) | 56* (34%) |
| | Anticipated products | 1 | 6 | 12 | 0 | 3 | 2 | 10 | 56 |
| | Unanticipated products** | 6* | 2 | 2 | 0 | 1 | 0 | 0 | 0 |
| Proximal off-target (outside protospacer) | # of alleles | 17* (36%) | 0 | 2* (6%) | 5* (22%) | 1 (4%) | 0 | 0 | N/A |
| | Anticipated products | 2 | 0 | 2 | 2 | 1 | 0 | 0 | |
| | Unanticipated products** | 20 | 0 | 0 | 6 | 0 | 0 | 0 | |
| Indels *** | | 2 (4%) | 11 (12%) | 10 (29%) | 1 (4%) | 5 (19%) | 0 | 0 | 0 |
| Anticipated on-target editing without any off-targets | | 17 (36%) | 54 (59%) | 10 (29%) | 17 (74%) | 17 (63%) | 6 (75%) | 29 (74%) | 110 (66%) |

Mice (and embryos) derived from zygotes injected with BE2, BE3, SaBE3, VQR-BE3, BE4, and ABE were analyzed. The edited alleles were analyzed based on location (on-target: within the 5-bp editing window of protospacer positions 4–8, counting the start of the PAM as position 21; bystander: within the protospacer; proximal off-target: outside, but within 200 bp, of the protospacer) and editing purity (anticipated products: C•G-to-T•A for CBE, and A•T-to G•C-for ABE; unanticipated products: any mutations other than anticipated products, including opposite-strand edits and indels). Single and double asterisk indicate that some alleles carried more than one mutation, or opposite strand editing, respectively. Triple asterisk indicates that alleles with indels also harbored additional mutations. Data from three ABE studies were analyzed
[a]ABE: this study
[b]ABE: ABE editing in the *Tyr* locus
[c]ABE: ABE editing in the *Ar* and *Hoxd13* locus. N/A, not analyzed

mutations while minimizing the production of opposite-strand edits, unanticipated mutations, and indels.

**Targeting mouse zygotes with cytosine base editors**. BE2 is a form of CBE that lacks nickase activity, and thus is expected to edit with lower efficiency, while minimizing indels and unanticipated mutations[9]. BE3 and alternative PAM variants such as VQR-BE3 and SaBE3 (which enable editing at sites with NGA and NNGRRT PAMs[22], respectively) contain nickase activity to increase the efficiency of base pair conversion after deamination. BE4 is an improved CBE containing a second uracil glycosylase inhibitor (UGI) domain and optimized linker architectures to reduce unanticipated products[23].

We co-injected mRNA encoding VQR-BE3 and a corresponding targeting sgRNA, and mRNA encoding BE4 and a targeting sgRNA, into mouse zygotes and obtained 24 founder mice. We identified in these mice 23 mutant alleles for VQR-BE3 and 27 mutant alleles for BE4 (Table 1, Supplementary Fig. 2a and b). Out of the 50 mutant alleles, 21 alleles for VQR-BE3 and 23 alleles for BE4 contained anticipated C•G-to-T•A mutations. Thirty-four alleles (17 alleles for VQR-BE3 and 17 alleles for BE4) exclusively carried on-target C•G-to-T•A substitutions and no additional mutations (Fig. 1b). Bystander editing outside the activity window but within the protospacer was observed in four alleles (15%) edited by BE4, and in none edited by VQR-BE3. Proximal off-target mutations were observed in six alleles (five alleles for VQR-BE3 and one allele for BE4), and deletions were

**Table 2 Summary of unanticipated products caused by various base editors**

| Edited alleles | CBE | | | | | ABE | | |
|---|---|---|---|---|---|---|---|---|
| | HF2-BE2 | BE3 | SaBE3 | VQR-BE3 | BE4 | ABE[a] | ABE[b] | ABE[c] |
| # of alleles* | 47 | 91 | 34 | 23 | 27 | 8 | 39 | 167 |
| # of alleles with unanticipated products* | 28 (60%) | 20 (22%) | 2 (6%) | 5 (22%) | 3 (11%) | 0 | 0 | 3 (2%) |
| # of Unanticipated products** | 40<br>20x G-to-A<br>6x C-to-A<br>7x C-to-G<br>5x G-to-T<br>1x G-to-C<br>1x A-to-G | 20<br>13x C-to-A<br>7x C-to-G | 2<br>2x C-to-A | 6<br>2x G-to-A<br>4x G-to-T | 3<br>2x C-to-A<br>1x T-to-C | 0 | 0 | 3<br>1x C-to-G<br>1x C-to-A<br>1x C-to-T |

Edited alleles were analyzed based for undesired nucleotide transition types. Single and double asterisk indicate that some alleles carried more than one mutation, or opposite strand editing, respectively

observed in six alleles (one allele for VQR-BE3 and five alleles for BE4). The product distribution of base editing following injection with VQR-BE3 or BE4 differed slightly. Unanticipated mutations were observed for five alleles (22%) following VQR-BE3 treatment, and for three alleles (11%) following BE4 treatment (Tables 1 and 2), consistent with the reduction of unanticipated editing events by the second UGI domain present in BE4 but not in BE3 variants[23]. While VQR-BE3 led to C-to-T transitions on both strands as well as to G-to-T transversions (which could arise from C-to-A transversions on the opposite strand), BE4 did not result in any observed opposite-strand editing. Notably, five alleles (10%) carried only proximal off-target mutations and/or deletions without any on-target mutations after VQR-BE3 or BE4 treatment (Tables 1 and 2, Supplementary Fig. 2a and b). Opposite strand editing, deletions, and transversions are consistent with error-prone DNA repair following uracil excision.

We also analyzed data from previously reported studies that treated mouse embryos with HF2-BE2 (a modified high-fidelity version of base editor 2)[14], BE3[18], and SaBE3[21] to introduce mutations into five distinct loci (Fig. 1b, Table 1, Supplementary Fig. 2c–e). On-target editing occurred in 68% (32 out of 47) of BE2-treated mutant alleles, in 85% (77 out of 91) of BE3-treated mutant alleles, and 59% (20 out of 34) of SaBE3-treated mutant alleles. Bystander editing was observed in 15% (seven alleles) for BE2, 9% (eight alleles) for BE3, and 38% (13 alleles) for SaBE3. Proximal off-target editing was identified in 36% (17 alleles) of BE2-treated zygotes and 6% (two alleles) of SaBE3-treated zygotes, but was notably absent among BE3-treated zygotes.

Next, we analyzed the distribution of products following HF-BE2, BE3, or SaBE3 treatment. BE3 resulted in higher product purity than HF2-BE2 and SaBE3, with anticipated on-target editing without any other mutations observed in 59% (54 out of 91) of alleles for BE3, versus 36% (17 out of 47) of alleles for HF2-BE2 and 29% (10 out of 34) of alleles for SaBE3. Unanticipated editing was observed in two instances out of 34 mutant alleles among SaBE3-treated zygotes, in 20 instances out of 91 mutant alleles among BE3-treated zygotes, and in 37 instances out of 47 mutant alleles among BE2-treated zygotes (Fig. 1c). Two alleles (4%) targeted by BE2, ten alleles (29%) targeted by SaBE3, and 11 alleles (12%) targeted by BE3 carried deletions, also consistent

with the lower expected indel frequency resulting from BE2's lack of nickase activity. These data suggest that BE3, VQR-BE3, and BE4 are comparably effective with ~60–75% of treated mouse embryos carrying only anticipated on-target mutations (Fig. 1d; Table 1). As expected, SaBE3 and BE4-treated zygotes showed higher product purity and fewer unanticipated (non-C•G-to-T•A) editing events than BE3, VQR-BE3, or HF2-BE2 (Fig. 1c; Table 1).

Although CBEs induced efficient C-to-T conversions at target sites within the editing window in 149 out of 222 total CBE-edited alleles, G-to-A mutations in 22 alleles were observed, consistent with CBE-induced C-to-T transitions on the opposite strand. Of note, opposite strand editing was only observed in the bystander and proximal off-target regions but not in the on-target region. It might arise from cytosine deamination, mediated either by the base editor or by endogenous cytidine deaminases, in exposed single-stranded regions of the opposite strand created by Cas9: guide RNA complexes[9]. Other cytosine substitutions, such as C-to-G, within protospacers are consistent with known DNA mismatch repair processes[23]. In agreement with previous reports that in most cases mutagenized cytosines outside the protospacer are preceded by a thymidine (5′-TC-3′), the preferred sequence context for APOBEC substrates[24,25], 44 out of 68 (65%) proximal off-target mutations observed in this analysis occurred at Cs that follow T. Mechanistic insight into base editing on both strands and proximal off-target base editing is limited since crystal structures of base editors have not yet been reported. Given the possibility of differences in DNA repair pathways across cell states and cell types, further studies are needed to unravel the mechanism and factors determining outcomes of base editing under various conditions, including in embryos.

**Targeting mouse genome with adenine base editors**. We investigated the editing location and product distribution of adenine base editing at two genomic loci in mouse zygotes generated in this study, as well as mouse zygote data from previously published experiments[19,21] (Fig. 1). We injected ABE mRNA and sgRNAs into mouse zygotes and obtained a total of eight founder mice carrying eight mutant alleles for ABE (Table 1 and Supplementary Fig. 3a). Among the on-target activity windows of the

eight alleles, all (100%) contained anticipated A-to-G mutations. Bystander editing within the protospacer was observed in two alleles (25%) edited at position 9, adjacent to the activity window for ABE. No proximal off-target mutations and deletions were observed. We also analyzed sequence data from 206 mutant alleles from founder mice and edited embryos[19,21] (Fig. 1, Supplementary Fig. 3b and c). Out of 39 alleles from the *Tyr* locus[19], 29 carried only on-target mutations and ten alleles carried bystander edits (A•T-to-G•C transitions) at protospacer positions 3 and 10 (Supplementary Fig. 3b). Out of 167 alleles from the *Ar* and *Hoxd13* genes[21], 110 carried only on-target mutations and 57 alleles carried additional A•T-to-G•C bystander edits at protospacer positions 3 and 9 (Supplementary Fig. 3c).

We also analyzed the product distribution of ABE. Remarkably, no proximal off-target mutations in the *Tyr* locus were observed, and no indels were found in the 214 alleles covering the *Wap*, *Csn*, *Tyr*, *Ar* and *Hoxd13* loci (Fig. 1, Table 1 and Supplementary Fig. 3b-c). Unanticipated mutations were detected in only three out of a total of 214 mutant alleles, at a C within the editing window of the *Ar* locus (Supplementary Fig. 3c). ABE editing from these studies showed similar overall product purity, with 65–75% of treated mouse embryos carrying only anticipated on-target mutations without any additional mutations (Fig. 1d and Table 1). These results together indicate that adenine base editing in mouse embryos proceeds with very high fidelity.

## Discussion

Since the majority of known human genetic variants associated with disease are point mutations[10,26], base editing has the potential to correct SNPs associated with many disorders in somatic cells and, possibly, the germline. As 61% of known pathogenic human SNPs can, in principle, be corrected by an A•T-to-G•C transition (ABE, 47% of pathogenic human SNPs) or by a C•G-to-T•A transition (CBE, 14% of pathogenic human SNPs)[10,26], both ABEs and CBEs could serve as appropriate tools to correct and study diseases with a genetic component. The observed ratios of desired:undesired products for CBE and ABE both (Fig. 1c; Table 2) compare very favorably to the typical outcomes of homology-directed repair (HDR), which are usually accompanied by an excess of indels[27–29]. Correcting certain genetic disorders, such as T cell lymphomas, will require the introduction of two or more mutations in the same homologous chromosome (linked mutations)[30,31]. This can be achieved using base editing[32] but is more challenging to accomplish using nucleases such as Cas9, which introduce deletions or translocations of sequences between multiple cleavage sites[3,7,33], and provoke undesired cellular responses to the presence of double-stranded breaks[7,8,34,35]. The remarkable rarity of proximal off-target mutations and the predominance of anticipated on-target and bystander edits in embryos exposed to ABE and to preferred CBE variants (Fig. 1b) has important implications in the use of these new genome editing tools for research and clinical applications.

## Methods

**Mice**. All animals were housed and handled according to the guidelines of the Animal Care and Use Committee (ACUC) of the NIH (https://oacu.oir.nih.gov) and all animal experiments were approved by the ACUC of National Institute of Diabetes and Digestive and Kidney Diseases (NIDDK) and performed under the NIDDK animal protocol K089-LGP-17. Base editing- and CRISPR-Cas9-targeted founder mice were generated using C57BL/6N mice (Charles River Laboratories) by the Transgenic Core of the National Heart, Lung, and Blood Institute (NHLBI).

**Targeted loci**. For adenine base editing, two transcription factor binding sites on chromosomes 5 and 11 were separately targeted by ABE and their corresponding sgRNAs. For cytosine base editing, transcription factor binding sites on enhancers

C and E between the *Csn2* and *Csn1s2b* genes were targeted simultaneously using VQR-BE3 and BE4, and their corresponding sgRNAs.

**mRNA preparation and microinjection into mouse zygotes**. The sgRNAs were designed using CRISPR.MIT.EDU, and subsequently produced using Thermo-Fisher's sgRNA In Vitro Transcription Service. ABE, BE4, and VQR-BE3 mRNAs were synthesized in vitro using the mMESSAGE mMACHINE T7 kit (Thermo-Fisher Scientific). Deaminase fused-Cas9 mRNA (50 ng/μl for each base editor) and sgRNAs (20 ng/μl for each sgRNA) were mixed and co-microinjected into the cytoplasm of fertilized eggs collected from superovulated C57BL/6N female mice (Charles River Laboratories) and implanted into oviducts of pseudopregnant fosters (Swiss Webster).

**Genotyping**. Genomic DNA was isolated from tail tissue, amplified by PCR, and followed by Sanger sequencing.

**Statistical analysis**. Chi-squared test was applied to compare ABE and CBE base editors using the categories, on-target, bystander, proximal off-target, and indels. Power analysis was done using a low effect size of 0.15 (based on the evaluated Cohen's d) and a significance level of 0.05. The $P$ values for the individual comparisons were subject for Bonferroni correction to correct for multiple testing. The analyses were done using R (https://www.R-project.org) and the packages 'pwr' (https://CRAN.R-project.org/package=pwr) and 'effsize' (https://CRAN.R-project.org/package=effsize).

## Data availability

The following publicly available DNA sequencing data are used in this study: (i) PMID 28244995 at SRP095234, (ii) PMID 29904106 at SRP140663, (iii) PMID 29702637 at PRJNA436188 and PRJNA436750 and (iv) PMID 29904106 at SRP140663.

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

## Acknowledgements

This study was supported by the IRPs of NIDDK and NHLBI. H.K.L. and S.K. were supported by a grant of the Korean Health Technology R&D Project, Ministry of Health & Welfare, Republic of Korea (HI15C1184 and HI17C2369, respectively). S.M.M. was supported by an NSF graduate fellowship. S.M.M. and D.R.L. acknowledge support from DARPA HR0011-17-2-0049, U.S. NIH RM1 HG009490, R01 EB022376, U01 AI142756, and R35 GM118062, and HHMI.

## Author contributions

Study design: H.K.L., D.R.L., C.L., L.H. Design of base editing studies and providing materials: S.M.M. and D.R.L. Generation of mutant mice: C.L. Experimental mouse work and data analysis: H.K.L. Computational analysis for sequence from an ABE. reference: S.K. Statistical analysis: H.K.L. and M.W. All authors contributed to writing the manuscript.

## Additional information

**Competing interests:** D.R.L. is a co-founder and consultant of Beam Therapeutics, a company using base editing to develop human therapeutics. The remaining authors declare no competing interests.

