## [Peer Review File · Nature Communications]

Reviewer #1

The manuscript by Lee et al. reports on the fidelity of various CBE and ABE base editors in mouse zygotes, based on the analysis of 436 mutant alleles. This large dataset is a valuable resource and guide for the future production of targeted mouse mutants as the degree of on-target, bystander and proximal off-target editing differs among the editors. The manuscript is well organized, easy to read and compares the author's results to previous literature.

Response

None required

Reviewer #2

Lee and colleagues perform an important and timely analysis of specificity of base editing using the constellation of such tools engineered by David Liu's lab. Of particular broad interest is the fact that this analysis is done in mice (where HDR efficiency may be lower).

Comments

1. The key data are presented in Table 1 and they are, frankly, very hard to digest. The authors need to come up with a way to represent these data such that one can derive some general conclusions about their findings by using a graphical display. For instance, one could take a generic (schematic) locus, and then, for each of the 6 base editors they analyse, graph out (perhaps as bars above the position of interest):

- Desired point mutation at the target
- Undesired point mutation at the target
- Any point mutation in the region covered by the gRNA
- All other point mutations in that haplotype

Response

We thank the reviewer for the suggestions, which were implemented in a new figure (figure 1) and the data are much easier to digest. The new graphs show (1) the percentage of on-target, bystander and proximal mutations as well as indels that are based on the locations of target sites, and (2) either desired (also called intended or anticipated) and undesired (also called unintended or unanticipated) mutations based on the editing purity.

2. Essentially this is the display in supplementary notes, except the nature of the alleles is, frankly, not important for the average user – the average user needs to know, what to expect from using a particular form of a base editor.

Response

We moved those data into Supplementary Figures.

3. The manuscript lacks any statistical analysis – which is essential. Given the number of alleles analyzed – what is the sensitivity of the data for all four types of alleles? In other words, what is the false negative rate in the data as presented?

Response

The question is not clear to us. Does the reviewer ask whether some of the alleles were incorrectly characterized (false negative and false positive)? All alleles were sequenced from both directions, i.e. both the Watson and the Crick strand were sequenced and all mutations have been clearly identified. We need more guidance on what the reviewer wants us to do.

4. Line 66: should be “within the on-target”

Response

We corrected the sentence. intended to show the total number of founders as well as the number of analyzed alleles (founders can carry two or more mutant alleles), but not

the number of edited alleles at on-target in the text. We edited the text in the manuscript and added the numbers.

5. "Product purity" is an awkward and ambiguous phrase (putting it mildly). "The intended allele was obtained in x%" should be used - or, for the Table - "intended allele."

Response

We believe that "product purity" is the appropriate term and reflects "intended mutation" or "anticipated mutation", which was now added to the text.

Thank you for handling our manuscript entitled “Targeting fidelity of adenine and cytosine base editors in mouse embryos”. We were pleased that the reviewers considered our work important and we thank them for their constructive comments. Specifically, based on the suggestions by reviewer #2, we added graphs (now figure 1) to illustrate the results in a comprehensive and clear way.

Reviewer #1

The manuscript by Lee et al. reports on the fidelity of various CBE and ABE base editors in mouse zygotes, based on the analysis of 436 mutant alleles. This large dataset is a valuable resource and guide for the future production of targeted mouse mutants as the degree of on-target, bystander and proximal off-target editing differs among the editors. The manuscript is well organized, easy to read and compares the author’s results to previous literature.

Response

None

Reviewer #2

Lee and colleagues perform an important and timely analysis of specificity of base editing using the constellation of such tools engineered by David Liu’s lab. Of particular broad interest is the fact that this analysis is done in mice (where HDR efficiency may be lower).

Comments

1. The key data are presented in Table 1 and they are, frankly, very hard to digest. The authors need to come up with a way to represent these data such that one can derive some general conclusions about their findings by using a graphical display. For instance,

one could take a generic (schematic) locus, and then, for each of the 6 base editors they analyse, graph out (perhaps as bars above the position of interest):

- Desired point mutation at the target
- Undesired point mutation at the target
- Any point mutation in the region covered by the gRNA
- All other point mutations in that haplotype

Response

We thank the reviewer for the suggestions, which were implemented in a new figure (figure 1) and the data are much easier to digest. The new graphs show (1) the percentage of on-target, bystander and proximal mutations as well as indels that are based on the locations of target sites, and (2) either desired (also called intended or anticipated) and undesired (also called unintended or unanticipated) mutations based on the editing purity.

2. Essentially this is the display in supplementary notes, except the nature of the alleles is, frankly, not important for the average user – the average user needs to know, what to expect from using a particular form of a base editor.

Response

We moved those data into Supplementary Figures.

3. The manuscript lacks any statistical analysis – which is essential. Given the number of alleles analyzed – what is the sensitivity of the data for all four types of alleles? In other words, what is the false negative rate in the data as presented?

Response

We have analyzed a total of 436 individual mutant alleles that were derived from injected mouse zygotes; 222 alleles had been edited by CBE variants and 214 alleles edited by ABE. This permitted us to determine the exact mutation structure for each individual allele.

We added statistical analyses in Supplementary Figure 1 and in the manuscript (underlined text). Supplementary Figure 1 shows additional analysis for the comparison of CBE and ABE in different categories of editing. The applied Chi-Squared test for categorical data was selected to calculate the significance between CBE and ABE. The test was applied on the allele numbers of the editing locations (Table 1, Supplementary Figure 1a) for on-target, bystander and proximal off-target mutations, as well as indels. The power for the Chi-Squared test was analyzed and resulted in an 83% probability that the test correctly rejects the null hypothesis, using a small effect size and a significance level of 0.05. Panels b-d in Supplementary Figure 1 show the base editing frequencies using boxplots for each categories of editing location. The individual *P-values* were calculated using a Chi-Squared test with subsequent Bonferroni corrected for multiple testing. Bystander editing (b) shows similar frequency for CBE and ABE ($P = 0.0012728$), but proximal off-targets ($P = 2.2932e-06$) (c) and indels ($P = 2.8952e-07$) (d) are highly significant between CBE and ABE. In addition, we added the number of mutations and edited alleles in Supplementary Figure 1e to provide a clear depiction that CBE introduces more often multiple bystander and proximal off-target mutations on the same allele than ABE.

4. Line 66: should be “within the on-target”

Response

We corrected the sentence. We intended to show the total number of founders as well as the number of analyzed alleles (founders can carry two or more mutant alleles), but not the number of edited alleles at on-target in the text. We edited the text in the manuscript and added the numbers.

5. "Product purity" is an awkward and ambiguous phrase (putting it mildly). "The intended allele was obtained in x%" should be used - or, for the Table - "intended allele."

Response

We believe that "product purity" is the appropriate term and reflects "intended mutation" or "anticipated mutation", which was now added to the text.

Reviewers' Comments:

Reviewer #2:

Remarks to the Author:

The revised manuscript adequately addresses concerns raised during initial review.